# Sugar inhibits brassinosteroid signaling by enhancing BIN2 phosphorylation of BZR1

**Zhenzhen Zhang**[1,2], **Ying Sun**[3], **Xue Jiang**[1], **Wenfei Wang**[1]*, **Zhi-Yong Wang**[2]*

**1** College of Life Sciences, Fujian Agriculture and Forestry University (FAFU), Fuzhou, China, **2** Department of Plant Biology, Carnegie Institution for Science, Stanford, California, United States of America, **3** Hebei Key Laboratory of Molecular and Cellular Biology, Key Laboratory of Molecular and Cellular Biology of Ministry of Education, College of Life Science, Hebei Normal University, Hebei Collaboration Innovation Center for Cell Signaling, Shijiazhuang, China

* wenfeiwang@fafu.edu.cn (WW); zywang24@stanford.edu (ZW)

## Abstract

Sugar, light, and hormones are major signals regulating plant growth and development, however, the interactions among these signals are not fully understood at the molecular level. Recent studies showed that sugar promotes hypocotyl elongation by activating the brassinosteroid (BR) signaling pathway after shifting Arabidopsis seedlings from light to extended darkness. Here, we show that sugar inhibits BR signaling in Arabidopsis seedlings grown under light. BR induction of hypocotyl elongation in seedlings grown under light is inhibited by increasing concentration of sucrose. The sugar inhibition of BR response is correlated with decreased effect of BR on the dephosphorylation of BZR1, the master transcription factor of the BR signaling pathway. This sugar effect is independent of the sugar sensors Hexokinase 1 (HXK1) and Target of Rapamycin (TOR), but requires the GSK3-like kinase Brassinosteroid-Insensitive 2 (BIN2), which is stabilized by sugar. Our study uncovers an inhibitory effect of sugar on BR signaling in plants grown under light, in contrast to its promotive effect in the dark. Such light-dependent sugar-BR crosstalk apparently contributes to optimal growth responses to photosynthate availability according to light-dark conditions.

**Data Availability Statement:** All relevant data are within the manuscript and its Supporting Information files.

**Funding:** This work was supported by grants from National Institute of Health (NIH, R01GM066258 to Z-Y.W., https://www.nigms.nih.gov), the National

## Author summary

Genetic studies of the brassinosteroid (BR) deficient mutants revealed its essential role in seedling development in the dark, but subsequent studies showed no significant difference in BR level between seedlings grown under light and darkness. We recently observed that light does affect BR levels in Arabidopsis, but in a sugar dependent manner. In the dark, sugar increases BR level as well as BR sensitivity by stabilizing the steroid response factor BZR1 through the Target of Rapamycin (TOR) signaling pathway. However, the BR level is decreased by sugar under light and by darkness on sugar-free medium. These observations raised the question of how the combinations of light and sugar modulate BR signaling. We addressed this question using genetic physiological analyses and found interestingly that sugar inhibits brassinosteroid response in light-grown plants by

Natural Science Foundation of China grant (NO. 31700254, http://www.nsfc.gov.cn)and FAFU-International Collaborative Program (KXb16005A, https://www.fafu.edu.cn) to W.W., the Postdoctoral Innovative Talent Support Program (BX201700052, http://www.chinapostdoctor.org.cn/index.html) and the China Postdoctoral Science Foundation (2018M642551, http://jj.chinapostdoctor.org.cn/website/index.html) to Z.Z. The funders had no role in study design, data collection and analysis, decision to publish, or preparation of the manuscript.

**Competing interests:** The authors have declared that no competing interests exist.

stabilizing the glycogen synthase kinase 3 homolog BIN2 and attenuating the dephosphorylation of BZR1, but independently of TOR. Our results indicate that sugar acts through distinct pathways to promote and inhibit BR signaling in dark and light conditions. Our work illustrates an intricate three-way crosstalk whereby the combination of light and sugar signals modulate the brassinosteroid signaling pathway to optimize growth according to both environmental and metabolic conditions.

## Introduction

Plant growth is highly sensitive to environmental light conditions, the levels of endogenous hormones, and the availability of photosynthates (sugars). Sugar not only provides essential material and energy for growth, but also functions as signaling molecules. The sugar signaling pathways mediate plant responses to starvation (low sugar) or excess of sugar, mostly through modulating hormonal pathways [1–5]. How light modulates the sugar-hormone interactions to optimize growth is not well understood.

Brassinosteroids (BR) are a major class of growth-promoting hormones that regulate a wide range of developmental and physiological processes, including photomorphogenesis. BR plays an essential role in plant developmental responses to darkness, so called skotomorphogenesis, as the BR-deficient mutants show strong de-etiolation or constitutive photomorphogenesis phenotypes in the dark [6]. While it was widely speculated that light would reduce BR levels to promote photomorphogenesis, experimental measurement showed surprisingly no significant difference in BR level between seedlings grown under light and those grown in the dark [7]. Further studies uncovered light-BR crosstalk through interactions between downstream components of the signaling pathways [8,9]. However, recent studies suggest that BR level and sensitivity are modulated by the combination of light and sugar conditions [2,10].

BR-responsive gene expression is mediated by the Brassinazole-Resistant 1 (BZR1) family transcription factors [11]. Both nuclear localization and DNA-binding activity of BZR1 are inhibited due to phosphorylation by the GSK3-like kinase Brassinosteroid-Insensitive 2 (BIN2) [12–16]. BR signaling through the BRI1 receptor kinase leads to inactivation and degradation of BIN2 [11,17,18], and dephosphorylation of BZR1 by protein phosphatase 2A (PP2A) [19]. Unphosphorylated BZR1 accumulates in the nucleus, where it recruits the TOPLESS family repressors to inhibit gene expression [20,21] and interacts with transcription factors of other hormonal and light signaling pathways to promote shoot cell elongation [8,22–24].

BZR1 protein level is regulated through several mechanisms. Sugar signaling through Target of Rapamycin (TOR) stabilizes BZR1. When seedlings are shifted from light to darkness and undergo starvation, BZR1 is degraded due to TOR inactivation [2,25]. The degradation of BZR1 and its homolog under starvation and stress conditions involves the autophagy pathway [2,26]. Phosphorylated BZR1 is degraded by the proteasome following PUB40-mediated ubiquitination in roots of Arabidopsis [27]. The dephosphorylated BES1, a homolog of BZR1, can be ubiquitinated and degraded by the SINAT E3 ligase in a light dependent manner [28]. In addition, BZR1 is also modified and stabilized by the small ubiquitin-like modifier (SUMO), and salt stress induces deSUMOylation of BZR1 to inhibit growth [29].

We recently observed that sugar decreases the BR level in Arabidopsis plants grown under light. However, after shifting light-grown Arabidopsis seedlings into darkness, the BR levels increased in seedlings grown on media containing sugar but decreased in those grown on sugar-free media, suggesting light-dependent effects of sugar on the BR pathway [2].

Therefore, we further tested how sugar affects BR responses under light conditions. We found that, in contrast to the positive effects of sugar on growth and BZR1 accumulation in the dark, high levels of sugar attenuated the BR promotion of hypocotyl elongation and the BR-induced BZR1 dephosphorylation in Arabidopsis grown under constant light. Further, sugar increased the level of BIN2. The inhibitory effects of sugar on hypocotyl elongation and BZR1 dephosphorylation are independent of the sugar sensors Hexokinase 1 (HXK1) and TOR. The results suggest that sugars act through distinct pathways to promote and inhibit BR signaling under different light-dark conditions. Such an intricate three-way crosstalk is likely important for optimizing growth according to both environmental condition and endogenous metabolic status.

## Results

### Sucrose inhibits BR-induced hypocotyl elongation in light

To test the effects of sugar on BR responses under light-grown conditions, we grew Arabidopsis seedlings on media containing various concentrations of sucrose with or without BR for five days under constant light. BR increased the hypocotyl lengths of seedlings grown on a sugar-free medium but had little effect on the seedlings grown on high concentrations (90 and 150 mM) of sucrose (Figs 1A and S1). These sucrose-dependent phenotypes were not caused by osmotic effects since the same concentrations of mannitol did not reduce BR's promotion of hypocotyl elongation (Fig 1B). Compared to plants expressing wild type BZR1-CFP, transgenic plants expressing bzr1-1D-CFP, a hypermorphic mutant form that is more effectively dephosphorylated by PP2A [12,19], had slightly shorter hypocotyls without BR treatment and longer hypocotyls after growth on BR-containing medium (Fig 1A and 1C), consistent with previous observations [30]. Sucrose had much weaker inhibitory effects on BR responsive

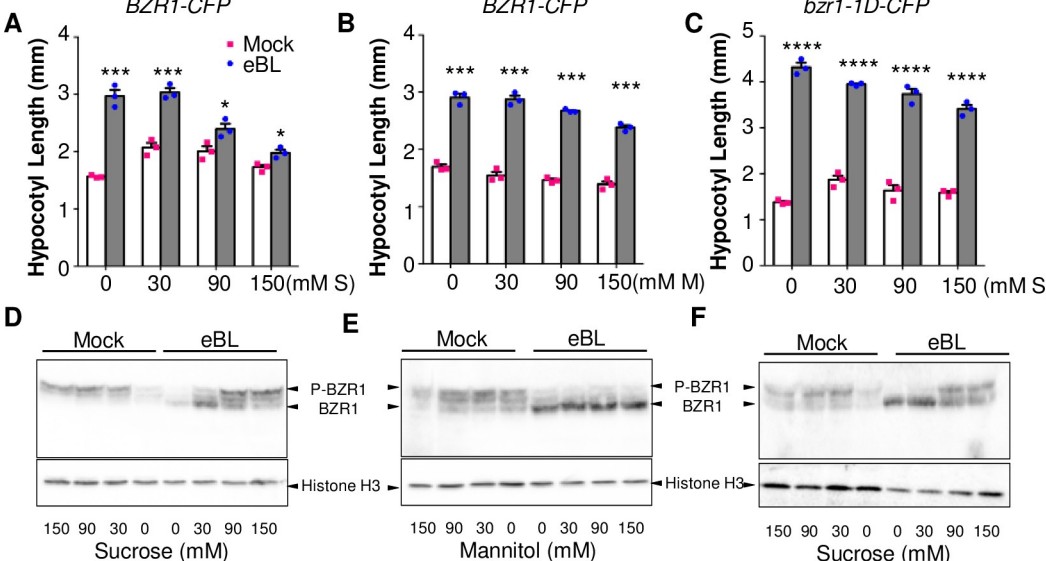

**Fig 1. Sucrose inhibits the BR-induced hypocotyl elongation and dephosphorylation of BZR1 in light.** (A-B) Hypocotyl length of transgenic Arabidopsis seedlings expressing BZR1-CFP (A and B) or bzr1-1D-CFP (C) grown on medium containing 100 nM epi-brassinolide (eBL) and various concentrations of sucrose (A and C) or mannitol (B). **** $P<0.0001$, *** $P< 0.001$, * $P<0.05$ (Student's t test). Error bars indicate the standard error of the mean (SEM, three replicates). (D-F) Immunoblot analysis of BZR1-CFP protein in the same batches of seedlings as in penal A to C. Histone H3 was probed as a loading control. P-BZR1 indicates phosphorylated BZR1; BZR1 indicates dephosphorylated BZR1. S, sucrose; M, mannitol.

hypocotyl elongation of the *bzr1-1D-CFP* than *BZR1-CFP* seedlings (Fig 1A and 1C). Taken together, the results suggest that sugar inhibits BR-induced hypocotyl elongation by inactivating BZR1.

## Sucrose attenuates BR-induced dephosphorylation of BZR1 in light

As BR promotes hypocotyl elongation through dephosphorylation of the BZR1 family transcription factors, we further examined whether sugar affects the phosphorylation status of BZR1. As shown in Fig 1D, exogenous BR treatment promoted dephosphorylation of BZR1 (as shown by the band shift in gel) in seedlings grown on mannitol or low concentrations of sugar (0 and 30 mM), but the effect on BZR1 dephosphorylation is decreased by high concentrations of sucrose (90 mM and 150 mM) (Figs 1D and 1E and S2). Furthermore, sucrose significantly decreased the BR effects on the expression of BZR1 target genes *CPD*, *DWF4* and *SAUR-AC* (S3 Fig), consistent with the effects on BZR1 phosphorylation. The bzr1-1D protein was more dephosphorylated than BZR1 under all conditions, although high concentrations of sugar also increased the phosphorylation of bzr1-1D (Fig 1F). These results indicate that sucrose inhibits BR-induced BZR1 dephosphorylation and activation in light-grown seedlings.

To further test whether sucrose decreases the sensitivity to BR, we treated seedlings grown on media containing 30 mM and 90 mM sucrose with BR for different times. BR caused more rapid dephosphorylation of BZR1 protein in seedlings grown under the low sucrose condition (30 mM) than the high sucrose (90 mM) condition (Fig 2). The total BZR1 protein level is lower on 30 mM sucrose media than on 90 mM sucrose (1:1.8) before BL treatment but

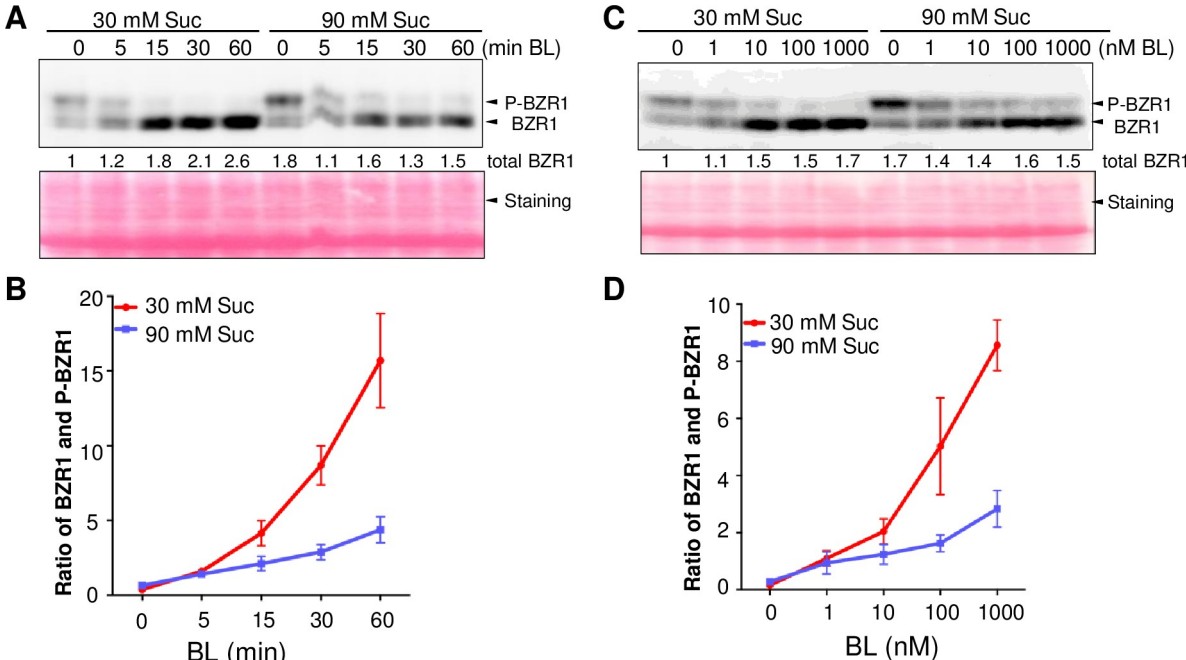

**Fig 2. High level of sucrose inhibits BR-induced dephosphorylation of BZR1.** (A) Immunoblot analysis of BZR1-CFP in seedlings grown in 30 mM or 90 mM sucrose medium after treatment with 1 μM brassinolide (BL) for the indicated time. Numbers below the image show the relative level of total BZR1 protein. (B) Quantification of the ratio between dephosphorylated and phosphorylated BZR1 (P-BZR1) using results shown in (A) and two additional biological repeats. (C) Immunoblot of BZR1-CFP in seedlings grown for five days on media containing 30 mM or 90 mM sucrose and different concentrations of BL after two-day germination in 30 mM sucrose medium. The relative levels of total BZR1 protein are shown below the image. (D) Quantification of the BZR1/P-BZR1 ratio using results shown in (C) and two additional biological replicate samples. Error bars indicate the SEM.

reached similar levels after 60 min BL treatments (Fig 2A), suggesting that the difference in phosphorylation is not due to a difference in BZR1 protein level. Base on three biological replicate results, the 15-minute BR treatments of seedlings grown in the low-sugar condition caused similar BZR1 dephosphorylation (ratio of dephosphorylated BZR1 to phospho-BZR1, BZR1/P-BZR1) to that caused by 60-minute BR treatments of seedlings grown in the high sugar condition (Fig 2B).

We further analyzed BZR1 in seedlings grown on media containing low (30 mM) or high (90 mM) concentrations of sucrose and various concentrations of brassinolide (BL). The results show that a higher BL concentration is required in high sugar condition than in low sugar condition to induce a similar increase of the BZR1/P-BZR1 ratio (Fig 2C and 2D). In the presence of BL (10 nM or higher), BZR1 was more dephosphorylated under low-sugar conditions than high-sugar conditions while the total BZR1 protein level remained similar. Collectively, these results demonstrate that high concentrations of sucrose inhibit BR signaling upstream of BZR1, specifically by inhibiting BR-induced BZR1 dephosphorylation, in light-grown seedlings.

## Sucrose suppresses BR signaling upstream of BIN2

Phosphorylation of BZR1 at Ser173 causes its binding and inhibition by the 14-3-3 proteins [16]. It has been reported that sugar increases the amount of many proteins that bind to the 14-3-3 proteins [31,32]. It was proposed that sugar may promote the binding of the 14-3-3 proteins to their target proteins and protect them from degradation by proteolysis [32]. Therefore, sugar may inhibit dephosphorylation or degradation of phospho-BZR1 by enhancing its binding to the 14-3-3 proteins. To test this hypothesis, we analyzed sugar effects on hypocotyl elongation and BZR1 dephosphorylation in transgenic plants that express the *BZR1^{S173A}–CFP* protein, which contains the S173A mutation that abolishes 14-3-3 binding [16]. The hypocotyl elongation induced by BR was inhibited by sucrose in the *BZR1^{S173A}–CFP* plants, and the BR-induced dephosphorylation of *BZR1^{S173A}* was also inhibited by sucrose, similar to wild-type BZR1 (S4 Fig). Thus, these results indicate that sucrose inhibits BR-induced dephosphorylation of BZR1 dephosphorylation and hypocotyl elongation independent of the 14-3-3 proteins.

BR induces BZR1 dephosphorylation by inactivating BIN2. We thus tested whether BIN2 mediates the phosphorylation of BZR1 on high sugar medium. It is reported that BIN2 interacts directly with BZR1 through a 12-amino acid BIN2-docking motif (DM) near the BZR1 C terminus. Deletion of this motif (bzr1-ΔDM) abolishes the interaction of BZR1 with BIN2 and prevents BIN2 phosphorylation of BZR1 *in vivo* [33]. Sugar was unable to cause phosphorylation of the bzr1-ΔDM protein (Fig 3A and 3B), suggesting that BIN2 mediates the sugar-promoted phosphorylation of BZR1. Further, the *bin2-triple* mutant with loss-of-function of BIN2 and its close homologs, *bin2bil1bil2*, showed reduced sensitivity to sugar (Fig 3C and 3D). In the presence of sugar, BR induced more dramatic hypocotyl elongation and BZR1 dephosphorylation in the *bin2bil1bil2* seedlings than wild type (Figs 3C and 3D and S5). These results suggest that sugar inhibition of BR responsiveness is dependent on BIN2 family proteins. Interestingly, we found that sugar increased the BIN2 level in the absence and presence of exogenous BR (Fig 3E). Together these results support a scenario that sugar causes accumulation of BIN2 protein to increase BZR1 phosphorylation and reduce BR responsive growth.

## Sucrose restrains BR signaling in light through unknown sugar signaling pathway

To determine which sugar-signaling pathway mediates the suppression on BR signaling in light, we carried out genetic tests of known sugar-signaling mutants. TOR and HXK1 are

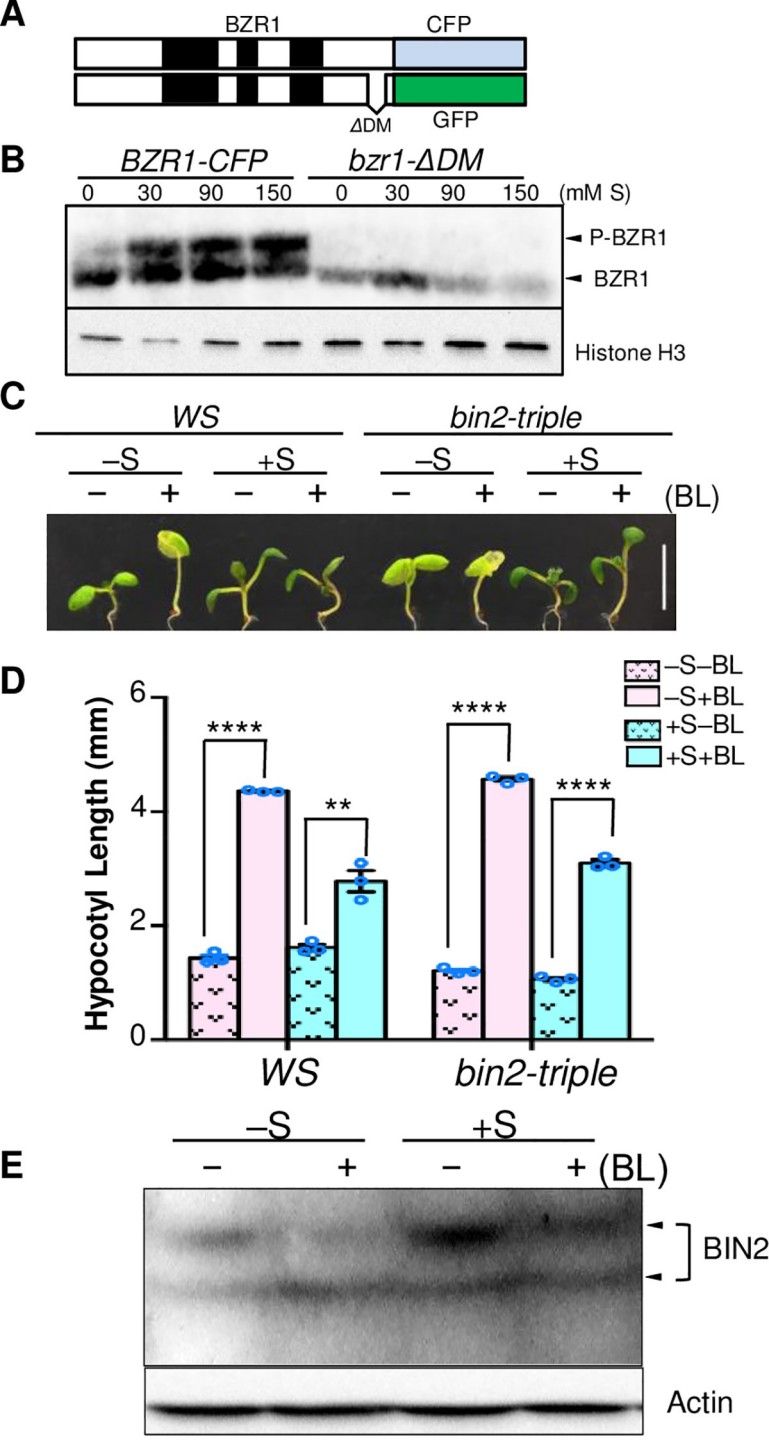

**Fig 3. Sucrose inhibition of BR signaling depends on the regulation of BIN2.** (A) Schematic presentation of BZR1-CFP and bzr1-ΔDM-GFP. (B) Immunoblot analysis of BZR1 protein in *BZR1-CFP*/Col-0 and *bzr1-ΔDM-GFP*/Col-0 seedlings grown on medium containing 100 nM eBL and different concentrations of sucrose (mM S). Histone H3 was probed as a loading control in the Immunoblot analysis. (C-D) Phenotypes and hypocotyl length of wild type (WS) and *bin2bil1bil2* (*bin2-triple*) mutant seedlings grown on media containing no sucrose (-S) or 90 mM sucrose (+S) and 0 (-BL) or 10 nM BL (+BL). Bar = 5 mm. **** $P<0.0001$, *** $P< 0.001$, ** $P<0.01$ (Student's t test). Error bars indicate the SEM (three replicates). (E) Anti-BIN2 immunoblot analysis of BIN2 proteins in seedlings grown on media containing 0 (-S) or 90 mM (+S) sucrose and 0 (-) or 10 nM (+) BL as indicated. The immunoblot was probed with an anti-Actin antibody as a loading control.

known to mediate plant growth response to moderate and high concentrations of sugars, respectively [25,34]. We thus performed sugar and BR treatments in the HXK1-deficient mutant *gin2-1* [25] and the estradiol-inducible TOR silencing line *tor-es* [34]. As HXK1 is the sensor of glucose, we grew seedlings with glucose and sucrose separately to distinguish their effect on BR signaling. The *gin2-1* mutant and wild type plants showed similar sugar inhibition of BR-promoted hypocotyl elongation (Fig 4A and 4B), unlike that *bzr1-1D* mutant, which is less inhibited by sugar compared to wild type (Fig 4A and 4B). In addition, the effects of BR and sugar on the BZR1 phosphorylation level were similar in the *gin2-1* background compared to wild type, with significant amounts of phosphorylated BZR1 remaining in the presence of both BR and high sugar (Fig 4C). The results suggest that HXK1 is not required for sugar inhibition of BR signaling.

Consistent with TOR promoting plant growth, when TOR is inactivated by estradiol-inducible RNAi suppression in the *tor-es* seedlings, seedlings are smaller compared to *tor-es* untreated with estradiol (Fig 4D). However, these *tor-es* plants showed similar sugar inhibition of BR signaling as wild type, based on their hypocotyl elongation and BZR1 dephosphorylation status in response to sugar and BR treatments (Fig 4D–4F). These results suggest that TOR is not involved in sugar suppression of BR signaling in light-grown plants.

## Discussion

Plant growth is highly sensitive to environmental signals and endogenous nutrient availability. Hormones, as internal growth regulators, are highly modulated by both environmental conditions and sugar availability to optimize growth and survival. Low sugar levels tend to be limiting for growth when plants are shaded or left in extended darkness, whereas surplus of photosynthate can also be inhibitory to growth. How plants deal with different sugar statuses under different environmental conditions is a key question relevant to crop yield.

We previously showed that sugar depletion in the dark causes BZR1 degradation and growth arrest; under such conditions, exogenous supply of sugar promotes BZR1 accumulation, thus enhancing BR promotion of shoot organ elongation [2,10]. Such sugar promotion of BR signaling in the dark is consistent with the need for maximum shoot elongation under shaded conditions while sugar is available, but arrest of such elongation when the sugar level is low. By contrast, we show in this study that for plants grown under full light, hypocotyl/shoot elongation is not a priority and a surplus of sugar is inhibitory to BR promotion of hypocotyl elongation. Such opposite effects of sugar on BR-dependent hypocotyl elongation are mediated by distinct mechanisms. In the dark, sugar increases BZR1 accumulation through TOR signaling. Under light, sugar increases BZR1 phosphorylation by increasing the level of BIN2 (Fig 5). It's worth noting that sugar also increases BR hormone accumulation in the dark but decreases BR level under light [2]. Thus, sugar has consistent effects on BR level and BR sensitivity, but these sugar effects are switched under dark and light conditions.

Genetic evidence suggests that distinct sugar signaling pathways are involved in the regulation of BZR1 in dark and light. While TOR mediates sugar-dependent stabilization of BZR1 in the dark [2] and BIN2 has been reported to be direct target of TOR-S6K signaling [3], inducible silencing of TOR made no obvious difference in the sugar effect on BZR1 phosphorylation or hypocotyl elongation responses to BR. TOR is inactivated when the sugar level is low, to trigger starvation response and growth arrest [34]. HXK1 is activated by high levels of glucose and mediate glucose inhibition of seedling growth and cotyledon greening [25]. There is also evidence showing that HXK1 mediates glucose positive regulation of BR signaling in promoting lateral root development [4]. It's somewhat surprising that HXK1 is also not required for

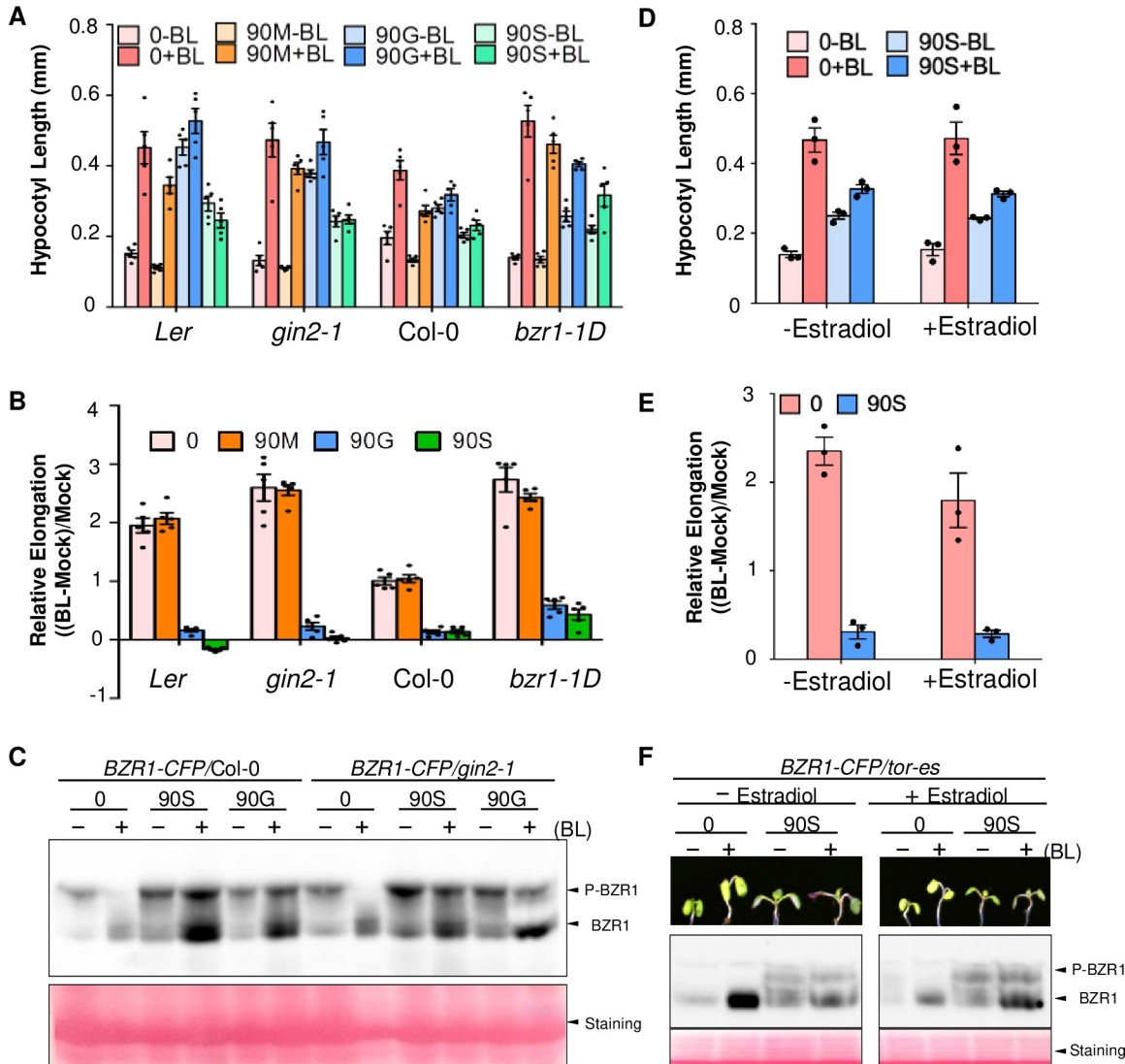

**Fig 4. Sucrose inhibits BR signaling independent of HXK1 and TOR.** (A) Hypocotyl lengths of seedlings grown on medium containing 0 mM or 90 mM of mannitol (90M), glucose (90G), or sucrose (90S) and no BL (-BL) or 100 nM BL (+BL). (B) Quantitation of BR sensitivity. Relative hypocotyl elongation is calculated as the ratio between the increase of length caused by BR and the length without BR treatment. Error bars indicate the SEM (five independent experiments, n≥ 20). (C) Immunoblot analysis of BZR1 protein in *BZR1-CFP*/Col-0 and *BZR1-CFP/gin2-1*, grown under conditions as shown in (A). (D) Hypocotyl lengths of *tor-es* seedlings grown on media containing 0 mM or 90 mM (90S) sucrose, 0 (-BL) or 100 nM BL (+BL), and 0 or 1 μM estradiol (+Estradiol). (E) Quantitation of BR sensitivity based on data in (D) using the method described for (B). Error bars indicate the SEM (three independent experiments, n≥ 20). (F) Images of seedlings and immunoblot analysis of BZR1 protein in seedlings described in (D).

sugar inhibition of BR-induced hypocotyl elongation and BZR1 dephosphorylation (Fig 4A–4C).

Alternative sugar signaling pathways may mediate inhibition of BR signaling. The SnRK1 pathway is inhibited by sugar but activated by sugar/energy deficiency, to deal with nutrient deficiency stress conditions [35]. SnRK1 is not known to be involved in responses to high sugar levels.

Recent studies show that glucose signaling is also mediated by Regulator of G-protein Signaling 1 (RGS1), a seven transmembrane guanosine-triphosphatase-activating protein that

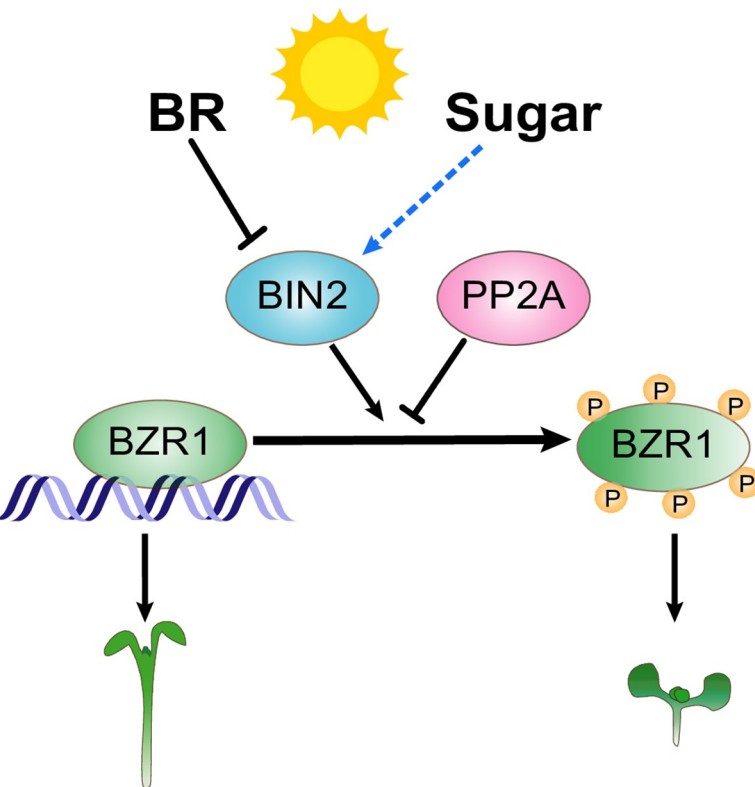

**Fig 5. A model of sugar suppressing BR signaling to modify plant growth in light.** In light, BR signaling inhibits BIN2 activity, which induces accumulation of active BZR1 to promote plant growth, whereas sugar induces the phosphorylation of BZR1 to suppress BR signaling probably through increasing BIN2 activity by unknown sugar signaling pathway. The black bars and arrows indicate previously reported mechanisms; the blue arrow indicates findings made in this study.

keeps GPA1 (Gα of G protein) in its inactive state [36–38]. Glucose-induced endocytosis of RGS1 releases its inhibition of G-protein self-activation which is triggered by phosphorylation at its C-terminal region by either the WNK kinases [38,39] or several LRR-RLKs [40–42]. Brassinosteroid insensitive 1 Like 3 (BRL3) and BRI1-associated Kinase 1 (BAK1) phosphorylates RGS1 promoting its endocytosis [41,42]. However, early seedling development and root growth of *rgs* mutants showed similar response to BL as wild type [41]. Recently, it was reported that glucose at low concentration increases the interaction between BRI1 and BAK1 in a manner dependent on BR biosynthesis [43], consistent with sugar increasing BR level in dark-treated seedlings [2]. Interestingly, high concentrations of glucose caused endocytosis of BRI1 and BAK1. Whether these cell surface receptors contribute to the sugar inhibition of BR signaling remains to be tested [43].

O-GlcNAc and O-fucose modifications have recently emerged as major sugar sensing mechanisms that impact on several hormonal pathways in plants [44,45]. Little is known about the interactions between these O-glycosylation pathways and BR signaling.

Our study indicates that light modulates sugar-BR crosstalk, however, the underlying mechanism remains unclear. Recently, several reports showed that light signaling inhibits BR signaling through photoreceptors regulating the activity of BZR1/BZR2 (BES1). Upon light activation, Phytochrome B, cryptochromes, and UVR8 bind with the dephosphorylated BZR1/BES1 and inhibit their DNA-binding activity [9,46–49]. CRY1 also interacts with BIN2 and enhances the interaction of BIN2 with BZR1 [47]. It would be very interesting to test in future

studies whether such direct interactions with phototransduction components rewire the cross-talk between sugar and BR pathways. The interactions between light and sugar pathways in modulating BR responses are important aspects of plant growth regulation that require further molecular investigation.

## Materials and methods

### Plant materials and growth conditions

Arabidopsis thaliana ecotype Columbia-0 (Col-0), *bzr1-1D* [12], *bzr1-1D-CFP*/Col-0, *BZR1-CFP*/Col-0 [30], *BZR1$^{S173A}$-CFP*/Col-0, Ler, *gin2-1* [25], *bzr1-ΔDM-GFP* [33], *BZR1-CFP*/*tor-es* [2] and *tor-es* [50] were all grown in a greenhouse with a 16-hr light/8-hr dark cycle at 22–24˚C for general growth and seed harvesting. All the plants were in Col-0 ecotype background except that *gin2-1* is in Landsberg erecta ecotype and *bin2,bil1,bil2* is in Wassilewskija (Ws).

### Sugar and BR Treatments and hypocotyl elongation assays

Seeds sterilized by 75% (v/v) ethanol were grown on solid 1/2 MS medium (pH 5.7) with 0.4% phytagel (Sigma) and 1% sucrose. After three days of incubation at 4˚C and two days of germination in light at 22–24˚C, seeds were transferred into 1/2 MS medium containing different concentrations of sugar and BR and were grown in continuous light for another 5 or 6 days as indicated. TOR silencing was induced in *tor-es* by adding 1 μM β-estradiol (Sigma, E8875) in the medium. Pictures were taken and the hypocotyl lengths were measured with the Image J. Raw data is shown in S1 Data. Seedlings were harvested for Western Blot.

### Primer design and Real-time quantitative PCR

Total RNA of above different sugar and BR treated seedlings was isolated by Spectrum Plant Total RNA Kit (Sigma-Aldrich, Shanghai, China). The primer sequences of BR responsive genes were listed in S1 Table. The qRT-PCR reactions were performed on QuantStudio 6 Flex Real-Time PCR System with TaKaRa Real-time qPCR Master Mix Kit. For each condition, the qRT-PCR experiments were performed with biological triplicates. Raw data is shown in S1 Data.

### Protein extraction and immunoblot analysis

For protein extraction, plants were frozen in liquid nitrogen, ground, weighed, and added into corresponding 2× SDS buffer (0.125 mM Tris-HCl [pH 6.8], 4% SDS, 20% Glycerol and 2% β-mercapto-ethanol). Samples were heated for 10 min at 95˚C, centrifuged at 10000g for 10 min, separated on a 7.5% (detecting BZR1 protein) or 15% (detecting Histone 3H protein) acrylamide gel and then blotted on PVDF membranes (Millipore, IPVH0010) in 192 mM glycine and 25 mM Tris-HCl with a Trans-blot Turbo blotting system (Bio-Rad) for 13 min. Membranes were blocked for 1 hour at room temperature in a blotting buffer (140 mM NaCl, 10 mM KCl, 8 mM Na2HPO4, 2 mM KH2PO4, 0.5% skim milk, and 0.1% Tween20, pH 7.4). The gel blots were incubated with the primary antibodies (anti-GFP, Transgene, HT801, at 1:1000; anti-BZR1 and anti-BIN2 (home-made) at 1 μg/ml; anti-Actin (Sigma, A2228) at 1:5000 dilutions; anti-Histone H3 (Sangon BBI antibody, AB51007) at 1:2000 dilutions). The secondary antibodies were used at 1:5000 dilutions for 1 hour.

## Supporting information

**S1 Fig. Sucrose inhibits the BR-induced hypocotyl elongation.** (A) Phenotypes of BZR1-CFP and *bzr1-1D*-CFP grown on media containing no sugar (-S), 30 to 150 mM sucrose (S) or mannitol (M), as in Fig 1. Bar = 5 mm. (B) Phenotypes of Col-0 grown on medium containing 0 (-BL) or 10 nM brassinolide (+BL) and 0 (-S) or 90 mM sucrose (+S) for six days. (C) Hypocotyl length of seedlings shown in panel B. Error bars indicate the standard error of the mean (SEM, three independent experiments, n≥ 20). **** $P<0.0001$ (Student's t test).
(TIF)

**S2 Fig. Sucrose inhibits the dephosphorylation of BZR1 protein induced by BR.** Immunoblot analysis of BZR1 protein with BZR1 antibody in Col-0 grown on medium containing 0 (-BL) or 10 nM brassinolide (+BL) and 0 (-S) or 90 mM sucrose (+S) for six days.
(TIF)

**S3 Fig. Sucrose inhibits BR responses of BZR1 target genes.** (A-C) Relative expression of *CPD*, *DWF4* or *SAUR-AC* analyzed by qRT-PCR in Col-0 seedlings grown on medium containing 0 (-BL) or 10 nM brassinolide (+BL) and 0 (-S) or 90 mM (+S) sucrose for six days. Error bars indicate the SEM (three independent experiments). ** $P< 0.01$, * $P<0.05$ (Student's t test).
(TIF)

**S4 Fig. Sucrose inhibits BR signaling independent of 14-3-3 protein.** (A) Hypocotyl length of *BZR1^{S173A}-CFP*/Col-0 grown on medium containing 100 nM eBL and indicated concentrations of sucrose. (B) Immunoblot analysis of BZR1^{S173A}-CFP protein in seedlings in (A). Histone H3 was probed as a loading control in the immunoblot analysis.
(TIF)

**S5 Fig. Loss of BIN2 function enhances BR-induced BZR1 dephosphorylation in the presence of sugar.** Immunoblot analysis of BZR1 protein in wild type and the *bin2-triple* mutant grown on media containing 90 mM sucrose and 0 (-BL) or 10 nM BL (+BL) for six days. The ratio of dephosphorylated BZR1 to phosphorylated BZR1 (BZR1/P-BZR1) was measured.
(TIF)

**S1 Table. List of primers used in this study.**
(XLSX)

**S1 Data. Raw data for all quantitative assays shown in the manuscript.**
(XLSX)

## Acknowledgments

We thank Dr. Jen Sheen for the mutant *gin2-1* and *tor-es*. We thank Marica Margis-Pinheiro and Tina Tingting Wang for editing the manuscript.

## Author Contributions

**Conceptualization:** Zhenzhen Zhang.

**Data curation:** Zhenzhen Zhang, Zhi-Yong Wang.

**Formal analysis:** Zhenzhen Zhang.

**Funding acquisition:** Zhenzhen Zhang, Wenfei Wang, Zhi-Yong Wang.

**Investigation:** Zhenzhen Zhang, Xue Jiang, Wenfei Wang, Zhi-Yong Wang.

**Methodology:** Zhenzhen Zhang, Wenfei Wang, Zhi-Yong Wang.

**Project administration:** Zhi-Yong Wang.

**Visualization:** Zhenzhen Zhang.

**Writing – original draft:** Zhenzhen Zhang, Zhi-Yong Wang.

**Writing – review & editing:** Zhenzhen Zhang, Ying Sun, Xue Jiang, Wenfei Wang, Zhi-Yong Wang.

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
