## [Decision Letter · Decision Letter 0]

3 Nov 2020

Dear Dr Wang,

Thank you very much for submitting your Research Article entitled 'Sugar inhibits brassinosteroid signaling by enhancing BIN2 phosphorylation of BZR1.' to PLOS Genetics. Your manuscript was fully evaluated at the editorial level and by two independent peer reviewers. The reviewers appreciated the attention to an important problem, but raised some substantial concerns about the current manuscript. Based on the reviews, we will not be able to accept this version of the manuscript, but we would be willing to review again a much-revised version. 

If you decide to revise the manuscript for further consideration at PLOS Genetics, please aim to resubmit within the next 60 days, unless it will take extra time to address the concerns of the reviewers, in which case we would appreciate an expected resubmission date by email to plosgenetics@plos.org.

[LINK]

We are sorry that we cannot be more positive about your manuscript at this stage. Please do not hesitate to contact us if you have any concerns or questions.

Yours sincerely,

Li-Jia Qu

Associate Editor

PLOS Genetics

Gregory P. Copenhaver

Editor-in-Chief

PLOS Genetics

Reviewer's Responses to Questions

**Comments to the Authors:**

Reviewer #1: The present manuscript describes that sugar inhibits brassinosteroid (BR) signaling pathway in the light, and BR induction of hypocotyl elongation in seedlings grown under light is inhibited by increasing concentration of sucrose. Through phenotypic analysis and immunoblot analysis, the authors demonstrate that sucrose inhibits the BR-induced hypocotyl elongation and dephosphorylation of BZR1 in light. Furthermore, the authors find that high sucrose inhibits BR-induced dephosphorylation of BZR1. Moreover, the authors present that sucrose inhibition of BR signaling in light dependents on the regulation of BIN2. This work illustrates an intricate three-way crosstalk whereby the combination of light and sugar signals modulate the BR signaling pathway to optimize growth according to both environmental and metabolic conditions, which is of novelty and interests to researchers in the fields of light, phytohormone and sugar signaling. However, the authors need to address the comments and concerns I raised below.

Major points:

1. The authors used BZR1-CFP and bzr1-1D-CFP to analyze the hypocotyl lengths (Figure 1), but did not include wild type or bzr1 /mutant or BZR1-RNAi plants. These data are not sufficient to support the authors’ conclusion that sucrose inhibits the BR-induced hypocotyl elongation and dephosphorylation of BZR1. The authors should perform these analyses again with at least wild type plants.

2. As shown in n Figures 2 A and C, as sucrose concentration increases, the protein levels of P-BZR1 and BZR1 increase accordingly, and P-BZR1 and BZR1 proteins are not at the similar levels at the very beginning. Therefore, it’s not appropriate to conclude that high sucrose inhibits BR-induced dephosphorylation of BZR1. Please clarify this complication.

3. The authors stated that high concentration of sucrose inhibits BR-induced dephosphorylation of BZR1. Unfortunately, the authors did not detect the protein levels of P-BZR1 and BZR1 in bin2-triple mutant, which is important to support their conclusion.

4. The authors conclude that sucrose inhibits BR signaling independent on 14-3-3 proteins, HXK1 and TOR. But the authors did not explain how high concentration of sucrose may inhibit BR signaling. Please discuss the possible molecular mechanisms. Moreover, the results showing that sucrose inhibition of BR signaling is independent of HXK1 and TOR should be included in the supplemental materials.

Minor points:

1. In Figures 1F and 3B, the loading control protein histone H3 is not at the same level.

2. In Figure S1B, “Mock” and “eBL” are not located at the central of the line. “Histone H3” does not show the band correctly. The similar problems are found in Figures 1, 2, 3, and 4.

3. The data shown in Figures 3C and 3D should be analyzed by student’s t test.

4. Given the authors’ conclusion that sucrose inhibits BR signaling independent of HXK1 and TOR, it appears that the trend of BZR1 phosphorylation in BZR1-CFP/Col-0 and the BZR1-CFP/gin2-1 seedlings shown in Figure 4C is not consistent with this conclusion. Please clarify.

Reviewer #2: Brassinosteroids (BRs) are a family of plant steroid hormones that play crucial roles throughout plant growth and development. Through BRI receptor, BR signal inhibits BIN2 kinase and results in dephosphorylation and activation of the master transcription factors BZR1 and BES1 to regulate thousands of BR target genes. In dark, BR promotes hypocotyl growth, while under light, BR inhibit hypocotyl elongation. It was shown that sugar-TOR signaling promote hypocotyl growth by stabilizing BZR1 protein when light grown seedlings shift to darkness (ref1). However, the mechanism of how BR inhibit seedling hypocotyl elongation is still unclear. The authors reported an interesting finding that under light high sucrose inhibits BR signaling by stabilizing BIN2 kinase and therefore increasing BZR1 phosphorylation and degradation/inactivation. This paper connects BR signaling with nutrient status and provides interesting observation that BIN2 and BZR1 protein levels correlate with sugar inhibition

Specific concerns

1. As hypocotyl length is one measurement of BR/BZR1/BES1 activity, it would be nice to have seedling pics of those various sugar & BL treatments, at least for Fig 1, 2,3. Reference 1 is a good example for combing phenotypes and quantitative measurements.

2. do authors also have other evidence for BZR1 activity, such as target gene expression change by Q-RT-PCR and binding of BZR1 to its target genes by ChIP-PCR? Reference 2 provides RT and ChIP-PCR target genes. The reason is that WB exposure varies from gel to gel, which makes it hard to compare between samples. For example, in fig 4 D and F, the measurements of hypocotyl length are not consistent with the BZR1 band intensity between samples treated w/ and w/o Estradiol.

3. BIN2 has been reported to be a direct target of TOR-S6K signaling (ref 3). When the authors claimed that the sugar inhibition is independent of TOR, the possibility was not ruled out that high sugar might reduce TOR activity, which releases BIN2 inhibition and results in decreased BZR1 activity.

Minor issues:

1. Fig 3E, the label needs to correspond to samples.

2. In some figures, such as Fig1D and E, western exposure time varies a lot that makes the similar samples look so different (for example, Suc 0, eBL Vs Mannitol 0, eBL)

Reference

1. Zhang Z, Zhu J Y, Roh J, et al. TOR signaling promotes accumulation of BZR1 to balance growth with carbon availability in Arabidopsis[J]. Current Biology, 2016, 26(14): 1854-1860.

2. Oh E, Zhu J Y, Wang Z Y. Interaction between BZR1 and PIF4 integrates brassinosteroid and environmental responses[J]. Nature cell biology, 2012, 14(8): 802-809.

3. Xiong F, Zhang R, Meng Z, et al. Brassinosteriod Insensitive 2 (BIN2) acts as a downstream effector of the Target of Rapamycin (TOR) signaling pathway to regulate photoautotrophic growth in Arabidopsis[J]. New Phytologist, 2017, 213(1): 233-249.

**Have all data underlying the figures and results presented in the manuscript been provided?**

Reviewer #1: Yes

Reviewer #2: Yes

PLOS authors have the option to publish the peer review history of their article (what does this mean?). If published, this will include your full peer review and any attached files.

Reviewer #1: No

Reviewer #2: No

---

## [Decision Letter · Decision Letter 1]

6 Apr 2021

Dear Dr Wang,

We are pleased to inform you that your manuscript entitled "Sugar inhibits brassinosteroid signaling by enhancing BIN2 phosphorylation of BZR1." has been editorially accepted for publication in PLOS Genetics. Congratulations!

Yours sincerely,

Li-Jia Qu

Section Editor: Plant Genetics

PLOS Genetics

Gregory Copenhaver

Editor-in-Chief

PLOS Genetics

Comments from the reviewers (if applicable):

Reviewer's Responses to Questions

**Comments to the Authors:**

Reviewer #1: The authors have addressed almost all of my comments and concerns, and I am satisfied with the revised manuscript.

**Have all data underlying the figures and results presented in the manuscript been provided?**

Reviewer #1: Yes

PLOS authors have the option to publish the peer review history of their article (what does this mean?). If published, this will include your full peer review and any attached files.

Reviewer #1: No

**Data Deposition**

http://datadryad.org/submit?journalID=pgenetics&manu=PGENETICS-D-20-01404R1

**Press Queries**

---

## [Editor Report · Acceptance letter]

12 May 2021

PGENETICS-D-20-01404R1 

Sugar inhibits brassinosteroid signaling by enhancing BIN2 phosphorylation of BZR1. 

Dear Dr Wang, 

We are pleased to inform you that your manuscript entitled "Sugar inhibits brassinosteroid signaling by enhancing BIN2 phosphorylation of BZR1." has been formally accepted for publication in PLOS Genetics! Your manuscript is now with our production department and you will be notified of the publication date in due course.

With kind regards,

Katalin Szabo

PLOS Genetics

On behalf of:
